# Effects of Oil Phase on the Inversion of Pickering Emulsions Stabilized by Palmitic Acid Decorated Silica Nanoparticles

Andrés González-González [1], Natalia Sánchez-Arribas [2] , Eva Santini [3], José Luis Rodríguez-Villafuerte [2], Carlo Carbone [2], Francesca Ravera [3] , Francisco Ortega [2,4], Libero Liggieri [3] , Ramón G. Rubio [2,*] and Eduardo Guzmán [2,4,*]

1 Departamento de Ingeniería Mecánica, Química y Diseño Industrial, Universidad Politécnica de Madrid, 28012 Madrid, Spain; aggrubio@hotmail.com
2 Departamento de Química Física, Universidad Complutense de Madrid, 28040 Madrid, Spain; natsanch@ucm.es (N.S.-A.); josero11@ucm.es (J.L.R.-V.); carlcarb@ucm.es (C.C.); fortega@quim.ucm.es (F.O.)
3 Istituto di Chimica della Materia Condensata e di Tecnologie per l'Energia-U.O.S. Genova, Consiglio Nazionale delle Ricerche, 16149 Genova, Italy; eva.santini@ge.icmate.cnr.it (E.S.); francesca.ravera@ge.icmate.cnr.it (F.R.); libero.liggieri@ge.icmate.cnr.it (L.L.)
4 Instituto Pluridisciplinar, Universidad Complutense de Madrid, 28040 Madrid, Spain
* Correspondence: rgrubio@quim.ucm.es (R.G.R.); eduardogs@quim.ucm.es (E.G.)

**Abstract:** Pickering emulsions stabilized by the interaction of palmitic acid (PA) and silica nanoparticles (SiNPs) at the water/oil interface have been studied using different alkane oil phases. The interaction of palmitic acid and SiNPs has a strong synergistic character in relation to the emulsion stabilization, leading to an enhanced emulsion stability in relation to that stabilized only by the fatty acid. This results from the formation of fatty acid-nanoparticle complexes driven by hydrogen bond interactions, which favor particle attachment at the fluid interface, creating a rigid armor that minimizes droplet coalescence. The comparison of emulsions obtained using different alkanes as the oil phase has shown that the hydrophobic mismatch between the length of the alkane chain and the C16 hydrophobic chain of PA determines the nature of the emulsions, with the solubility of the fatty acid in the oil phase being a very important driving force governing the appearance of phase inversion.

**Keywords:** emulsions; fatty acid; hydrophobic mismatch; phase inversion; Pickering; silica nanoparticles

## 1. Introduction

The stabilization of interfaces as result of the adsorption of nano- and microparticles has received increasing interest over the years [1–4]. This is due to the importance of particle-laden interfaces on the fabrication of many systems of technological and industrial interest, including emulsions, thin films, foams or colloidosomes, which are involved in different areas such as drug delivery, renewable energies, cosmetics, froth flotation, ink-jet printing, fabrication of porous materials and foams, or functional foods [5–7].

Among the systems stabilized by particle-laden interfaces there is no doubt that Pickering emulsions are probably among the most studied [8–13]. Nevertheless, several aspects related to these systems are still not fully understood, e.g., the transport of particles from the bulk to the interface [14], the role of the rheology and dynamics of the particle-laden interface on the emulsion stability [15], or the role of the particle wettability on their attachment to the interface and on the emulsion stabilization [16,17]. The latter is also related to the phase diagram of Pickering emulsions, i.e., the control of the nature of the continuous and dispersed phases in the emulsions. This is particularly important because the nature of the two phases can be critical for applications in which release from the emulsions of compounds with poor availability and low solubility is required [18].

The particle wettability for the interface defines different regions in the phase diagram depending on the nature of the continuous and dispersed phases: oil in water (o/w) and water in oil (w/o) emulsions [19]. The modification of particle wettability can be attained by chemical reaction, e.g., silanization of silica surfaces, thiolization of gold surfaces, or simply by the physical attachment of surface-active molecules through different type of interactions, e.g., electrostatic forces, hydrogen bonds, etc. [3,16,17,19–21]. For simplicity we have focused our interest on the last approach for trying to drive Pickering emulsions through phase inversion. For silicon dioxide particles, the physical adsorption of surface-active molecules allow one to define three different regions for the wettability of particles by the water/oil interface. In two of them the hydrophilicity (low contact angle) of the particles is high, leading to them remaining preferentially in the aqueous phase resulting in the formation of oil in water emulsions (o/w). In the third one the particles become hydrophobic (high contact angle) and remain preferentially in the oil phase, allowing the stabilization of water in oil emulsions (w/o) [17,19,22]. Even though, the role of surfactant concentration in the double inversion phenomenon is well-known [17,19,23], less attention has been paid to the role of the interaction of the hydrophobic tail of the surfactant with the oil molecules in the phase behavior of Pickering emulsions. This work addresses such a problem by studying the role of the affinity of the hydrophobic chain of a fatty acid (palmitic acid) and the alkyl chain of the oil phase on the stabilization of the Pickering emulsions. Contrary to what has been most frequently examined [17,19], in this work the functionalization of the particles occurs in situ at the water/alkane interface [24], thus the modification of the wettability of the particles takes place only after the droplet/continuous phase interface is formed. Furthermore, particle functionalization is controlled using non-electrostatic interactions, in particular hydrogen bonds between the non-dissociated silanol groups at the surface of the silica nanoparticles (SiNPs) suspended in the aqueous phase and the carboxylic head group of the palmitic acid (PA) molecules dissolved in the oil phase.

## 2. Materials and Methods

### 2.1. Chemicals

Palmitic acid (PA) was purchased from Sigma–Aldrich (Saint Louis, MO, USA) with purity higher than 99%. Different alkanes were used for the preparation of the PA solutions (see Table 1).

**Table 1.** List of alkanes used as oil phase in this work.

| Alkane | Purity | Supplier |
| --- | --- | --- |
| *n*-Hexane | >95% | Merck (Darmstadt, Germany) |
| *n*-Decane | ≥95% | Sigma-Aldrich (Saint Louis, MO, USA) |
| *n*-Dodecane | >95% | Acros Organics (Geel, Belgium) |
| *n*-Tetradecane | ≥99% | Sigma-Aldrich (Saint Louis, MO, USA) |

Hydrophilic silica nanoparticle dispersions containing 1% w/w of solid content were made by dilution of a commercial dispersion of Ludox® HS-40 colloidal silica with 40% w/w of solid content (Sigma-Aldrich, Saint Louis, MO, USA). Ludox® HS-40 is a colloidal dispersion of negatively charged particles with a $\zeta$-potential of $-35.7 \pm 0.8$ mV, as measured using a Zetasizer Nano ZS (Malvern Instrument Ltd., Malvern, UK) technique, and have a pH around 9.8. The apparent hydrodynamic diameter as was determined using a Zetasizer Nano ZS (Malvern Instrument Ltd., Malvern, UK) dynamic light scattering is about $30 \pm 4$ nm and the specific surface area of 220 m$^2$/g according to the supplier.

Ultrapure deionized water used for cleaning and solution preparation was obtained by a multicartridge purification system aquaMAX$^{TM}$-Ultra 370 Series (Young Lin Instrument, Co., Anyang, Korea). The water used had a resistivity higher than 18 M$\Omega \cdot$cm, and a total organic content lower than 6 ppm.

## 2.2. Emulsion Preparation and Characterization

Emulsions with a total volume of 10 mL were prepared using a high-power laboratory mixer (IKA ULTRA TURRAX T25, Staufen, Germany) operating at 10,000 rpm for 10 min. These emulsions contained different volume fractions of the oil and water phases. The oil volume fraction is defined as,

$$\phi_o = \frac{V_o}{V_o + V_w},\tag{1}$$

and the water volume fraction as

$$\phi_w = 1 - \phi_o,\tag{2}$$

where $V_o$ and $V_w$ are the volumes of oil and aqueous phases in the emulsion, respectively. The oil phase of emulsions was a PA solution in alkane with concentration ($c$) varied in the 0.02–100 mM range. For all emulsions, aqueous dispersions containing 1% w/w of silica nanoparticles were used as aqueous phase. The results reported here focus on the stability of the emulsified fraction, quantified by the evolution of the height of the emulsion column.

Optical microscopy images of emulsions were obtained using a Nikon Eclipse 80i, (Nikon Inc., Minato, Japan) microscope coupled to a CCD camera (model C8800-21C, Hamamatsu Corp., Nakaku, Japan). The average diameter of the drops was obtained using ImageJ, averaging over a set of 10 images, containing at least 25 droplets per image.

## 3. Results

### 3.1. Effect of SiNPs-PA Interaction on the Emulsion Stabilization

Figure 1a shows a set of images representing the stability of emulsions stabilized by bare silica nanoparticles, PA and between SiNPs-PA at the water/decane interface after 24 h of aging. The first conclusion extracted from such images is that silica particles alone do not stabilize the emulsions. This is due to the high hydrophilicity of the particles which hinders their trapping at the water/decane interface. Furthermore, the stabilization of emulsions only using PA is not effective enough, and the small emulsion column remaining after 24 h of aging is not enough to guarantee the long-term persistence of such emulsions. On the contrary, when PA and silica particles are combined, a significant increase in the height of the emulsion column is found. This results from the synergetic effect in relation to emulsion stabilization of the interaction between SiNPs and PA at the water/alkane interface which leads to the formation of hydrophobic particle-fatty acid complexes adsorbed at the water/oil interface, leading to a long-term stability, contrary to emulsions stabilized exclusively by fatty acid molecules. In fact, no significant signatures of destabilization are observed for the Pickering emulsion after 45 days of aging (see Figure 1b). The results confirm that the interaction of SiNPs and PA at the water/alkane interface present a synergetic effect in relation to emulsion stabilization, in agreement with previous results by Santini et al. [25], providing a larger long-term stability against coalescence in relation to that of the emulsions stabilized only by fatty acid molecules. In fact, it has been observed that the time needed for the emulsion to reach a height equal to a half of the total liquid volume, $t_{1/2}$, is about 5 h for the conventional emulsion, whereas for the Pickering emulsion it is 1 day longer. This enhancement of the stability of the emulsions associated with the presence of nanoparticles can be understood by considering that the interaction of particles and PA through hydrogen bonds, occurring simultaneously to drop formation, is an efficient process to obtain fatty acid decorated particles. These modified particles can adsorb onto the drop surface leading to the formation of a protective shell surrounding the drops, able to prevent drop coalescence and consequently, to enhance the emulsion stability. It should be noted that the synergism between SiNPs and PA molecules is also observed from surface tension data. This is evidence that the surface tension of the mixed system always appears below that corresponding to PA solutions with the same concentration, and hence the PA-SiNPs interaction presents a synergistic effect on the reduction in the water/alkane interfacial tension [25].

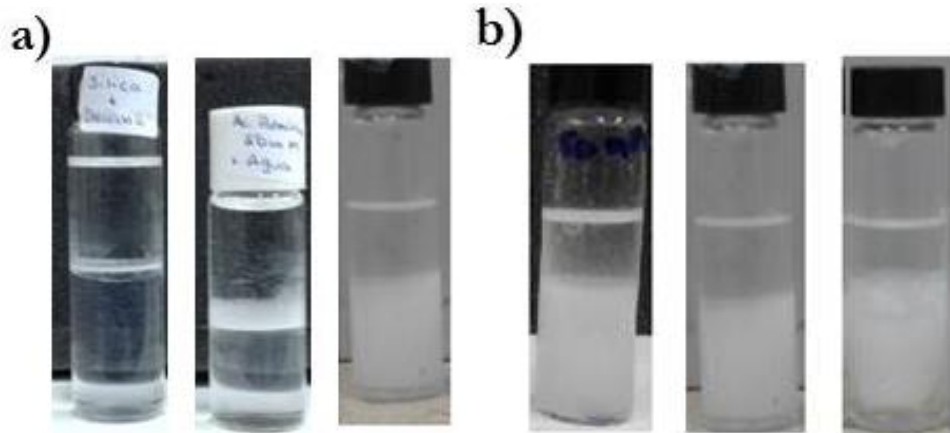

**Figure 1.** (**a**) Set of images representing emulsions ($\phi_o$ = 0.5) obtained at the water/decane interface using different stabilizing agents (from left to right): silica nanoparticles (1 wt% in particle content), palmitic acid (*c* = 50 mM) and palmitic acid (*c* = 50 mM) + silica nanoparticles (1 wt% in particle content). (**b**) Set of images representing the aging process of a Pickering emulsion ($\phi_o$ = 0.5) obtained at the water/decane interface stabilized using palmitic acid (*c* = 50 mM) + silica nanoparticles (1% w/w in particle content) (from left to right): emulsion just after preparation, emulsion after 1 day of aging and emulsion after 45 days of aging.

This work is focused on the role of the nature of the alkane in the phase inversion of Pickering emulsions, thus no further comparisons between Pickering emulsions and conventional ones stabilized only by the fatty acids will be included. A detailed discussion of this issue was presented by Santini et al. [25] in a previous publication.

### 3.2. Pickering Emulsions Stabilized by the Interaction of SiNPs and PA at the Water/Oil Interface: The Case of the Water/Decane Interface

The volume fraction of the alkane phase $\phi_o$ plays a central role in emulsion stabilization because it governs the ratio between the number of PA molecules available for the interaction with SiNPs (PA molecules/SiNPs ratio), thus affecting to the wettability of particles for the fluid interface. Therefore, the first step, once the emulsions were prepared, is to classify them according to their nature.

The emulsion type was evaluated by dispersing small fractions of freshly prepared emulsions (t = 0 h, i.e., just after stopping shearing) in vials containing a large volume of the pure liquids constituting the emulsion phases, i.e., water and alkane. When a small fraction of o/w emulsion is added to water, the emulsion is perfectly dispersed in the liquid due to the similarity between the water and the continuous phase of the emulsion. However, if the aliquot of o/w is added to decane, the emulsion settled down to the bottom of the vial containing decane. In the case of w/o emulsions, the phenomenology is just the opposite of what was obtained for o/w emulsions. For the sake of example, Figure 2 shows the behavior obtained in the dispersion of o/w and w/o emulsions in vials containing water and decane.

The dispersion tests point out the different affinity of the emulsions to a specific medium depending on the nature of their continuous and dispersed phases, with the nature of the continuous phase determining the emulsion dispersion in pure liquids. The dispersion tests made it possible to establish the nature of the different emulsions, which provides the basis for drawing a phase diagram of the Pickering emulsions stabilized by PA-decorated SiNPs having water and decane as fluid phases. It should be noted that the dispersion tests provide mainly information about the nature of the continuous phase, helping to obtain a first idea of the nature of the obtained emulsions. However, the dispersion tests do not provide information related to the internal structure of the droplets, making it difficult to identify the formation of multiple emulsions. In our case, the

formation of multiple emulsions can be ruled out by optical microscopy images (see below).

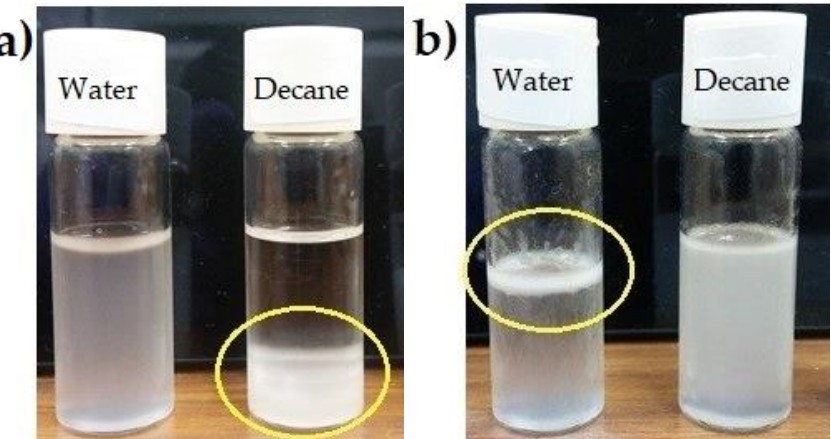

**Figure 2.** Dispersion tests in water and decane for o/w ($c = 20$ mM) (**a**) and w/o emulsions ($c = 70$ mM) (**b**) with $\phi_o = 0.3$ and a fixed particle concentration in the aqueous phase (1% w/w). The regions marked with the yellow ellipses indicate the emergence of phase separation in the dispersion test.

Figure 3 shows a simplified two-dimensional phase diagram obtained for the studied system as a function of the decane ($\phi_o$) or water ($\phi_w$) volume fractions and the initial concentration of palmitic acid ($c$) in the oil phase. This plot allows one to define the different emulsification regions, i.e., o/w and w/o, as well as the no emulsification one.

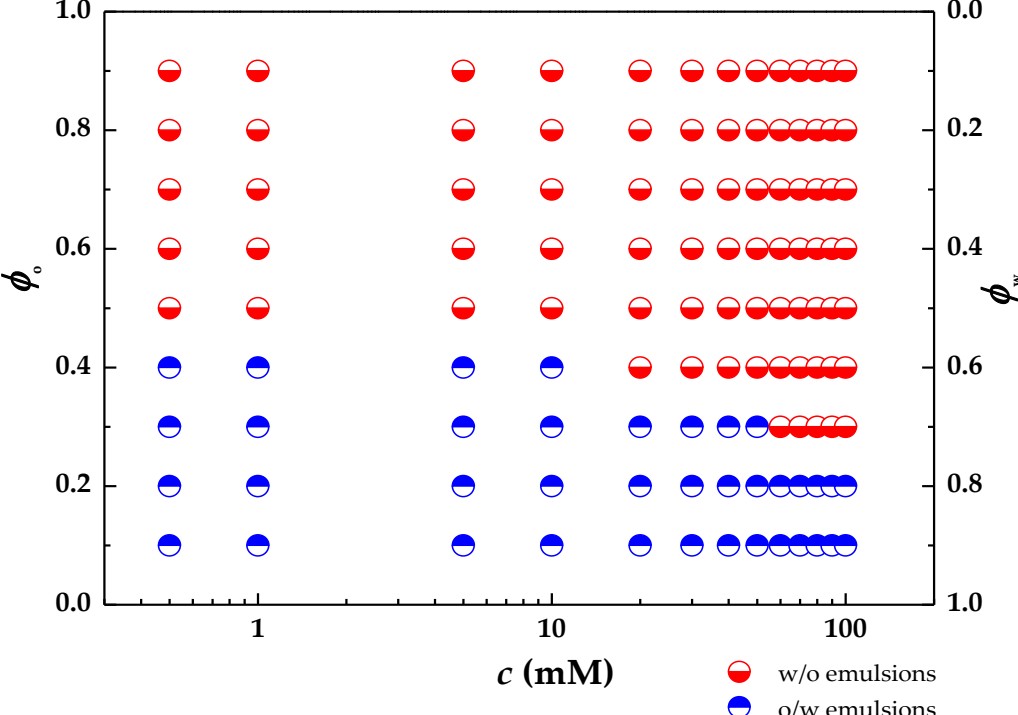

**Figure 3.** Simplified phase diagram for emulsions (water-decane system) stabilized by PA-decorated SiNPs represented as a function of the palmitic acid concentration $c$ and the volume fractions of oil ($\phi_o$) and water ($\phi_w$) phases. The different regions in which o/w and w/o emulsions appear are showed with different type of symbols. The emulsions were stabilized by a fixed concentration of particles in the aqueous phase of 1% w/w.

The phase diagram evidences the important role of the PA concentration in the control of the emulsion type. This can be explained by considering its impact on the degree of hydrophobicity of the NPs, and hence its modification allows tuning of both the emulsion type and its stability. Indeed, the different affinity of the decorated particles for the water/decane interface defines different contact angles by the interface, modifying the droplet curvature [26,27]. The results showed that both o/w and w/o emulsions can be obtained by varying the palmitic acid concentration for $\phi_o$ values between 0.3 and 0.4, whereas for $\phi_o$ below 0.3 and above 0.4, only o/w and w/o emulsions were obtained, respectively.

The more favored formation of o/w emulsions for the lowest volume fractions of decane can be understood by considering that the decrease in the oil volume fraction reduces the number of palmitic acid molecules available for modifying the nanoparticle hydrophobicity. Thus, the particle hydrophobicity trends to minimum values, which forces them to remain mostly immersed in the aqueous phase. This favors the stabilization of o/w emulsions. A similar explanation can be used for the formation of w/o emulsions at the highest values of volume fraction of decane. Thus, the increase in the alkane volume fraction increases the number of palmitic acid molecules available to change the hydrophobicity of the particles, which pushes the system to a phase inversion, resulting in the formation of w/o emulsions. For alkane volume fractions equal to or greater than 0.5, the formation of w/o type emulsions is found to be exclusive. This may be interpreted by considering that the number of fatty acid molecules is high enough to confer a significant modification in the degree of hydrophobicity of the nanoparticles. The results provide evidence that the PA molecules/SiNPs number ratio plays a central role for the stabilization and nature of the emulsions. This is clear considering that the higher the palmitic acid concentration the lower the volume fraction for the formation of o/w emulsions.

The above results point out that the presence of palmitic acid allows modulation of the nature of the emulsions for a given decane volume fraction (in the range 0.3–0.4) leading in some cases to the appearance of transitional phase inversions. This inversion in the type of emulsion phase is governed by the change of the hydrophilic-lipophilic balance (HLB), of the stabilizing agents, i.e., their affinity for the different phases involved in the interface formation. A similar transition was reported by Binks and Lumsdon [28] in surfactant-stabilized emulsions, in which the change in the type of emulsion is the result of the addition of a co-surfactant, whereas in this work the phase inversion occurs as a result of the increase in the number of molecules available for the modification of the hydrophilic silicon dioxide particles. It is worth mentioning that in this work, the number of palmitic acid molecules was not found to be enough to push the system through a phase second inversion, i.e., the double inversion phenomenon. This requires that the concentration of palmitic acid increases above a threshold value allowing the formation of palmitic acid bilayers on the surface of the silicon dioxide nanoparticles, which results in the rehydrophilization of the palmitic acid-decorated nanoparticles and favors again the formation of o/w [29]. Figure 4 shows a set of images allowing a clear evaluation of the effect of the PA molecules/SiNPs number ratio on the phase transition of the studied Pickering emulsions.

Together with the transitional phase inversion mediated by the change of the HLB of the particles as the palmitic concentration is increased, a second type of phase transition appears by varying the volume fraction of the liquid phases at fixed palmitic acid concentrations. This is the so-called catastrophic phase inversion [30], which drives the emulsions from o/w type to a w/o type as the volume fraction of oil is increased, emerging again as result of the change of the PA molecules/SiNPs number ratio in the system.

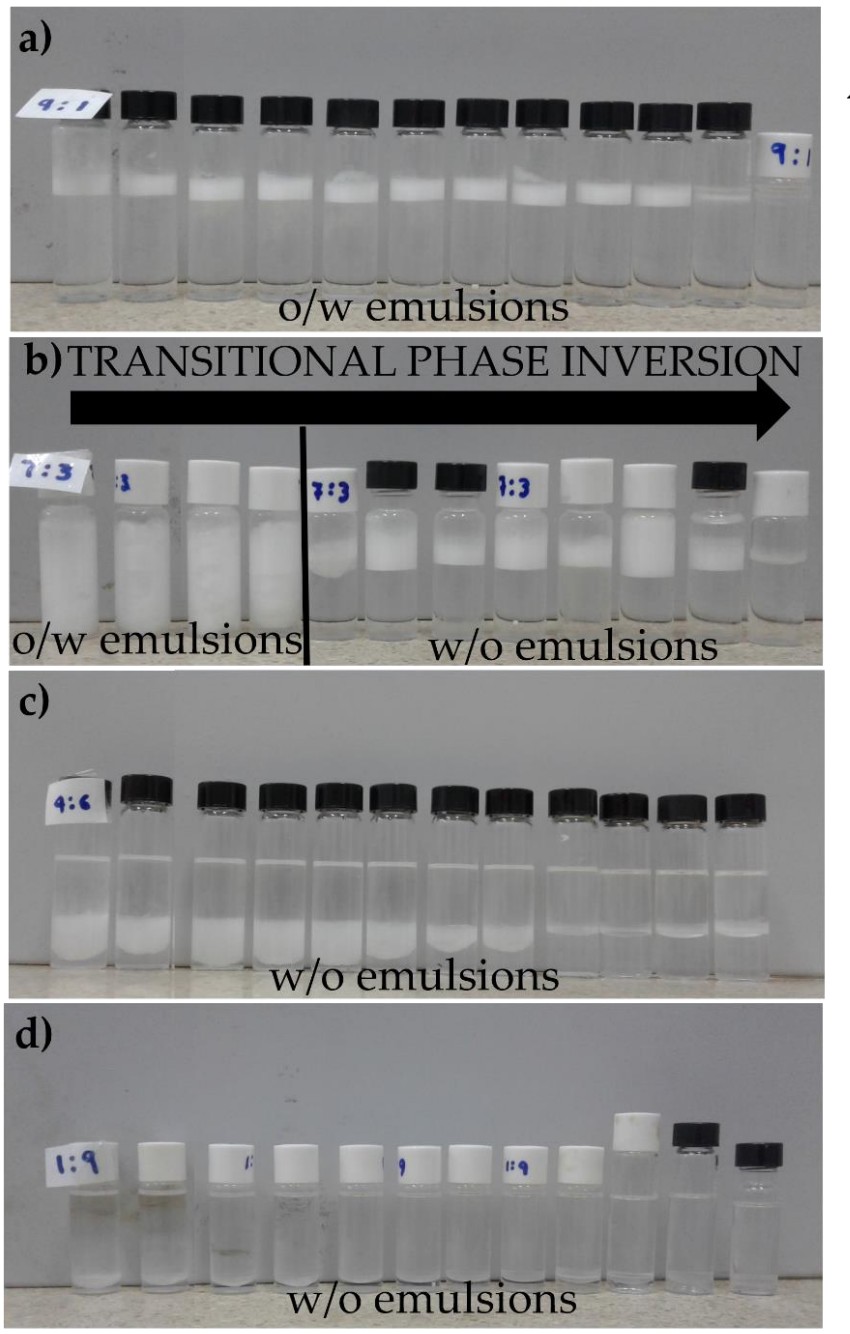

**Figure 4.** Set of images for Pickering emulsions prepared with different volume fractions of oil phase and palmitic concentrations (from left to right in all the panels the vials contain emulsions with palmitic acid concentrations (mM) in the oil phase: 100, 90, 80, 70, 60, 50, 40, 30, 20, 10, 5 and 1). (**a**) $\phi_{O}$ = 0.1. (**b**) $\phi_{O}$ = 0.3. (**c**) $\phi_{O}$ = 0.6. (**d**) $\phi_{O}$ = 0.9. The different types of phase inversion and emulsions are also indicated in the figure. The emulsions were stabilized by a fixed concentration of particles in the aqueous phase of 1% w/w. The images correspond to emulsions after one month of aging.

It should be noted that the PA molecules/SiNPs number ratio also affects the stability of the obtained emulsions, evaluated as the height of the remaining emulsified column. Thus, the decrease in the oil volume fraction can provoke the emulsion destabilization as a result of the decrease in the number of palmitic acid molecules available and the increase in the silicon dioxide nanoparticle number. In fact, the decrease in the oil volume fraction decreases the PA molecules/SiNPs number ratio, which leads to a decrease in the maximum degree of hydrophobization that could be reached by the particles. This hinders

the attachment of the particles onto the drop surface, preventing the emulsion stabilization as is evidenced from the small height of the emulsion column obtained when $\phi_o = 0.1$. Following a similar idea, it is possible to justify the destabilization of emulsions for the highest values of $\phi_o$, in which the PA molecules/SiNPs number ratio is relatively high, thus favoring the formation of palmitic acid bilayers on the SiNPs surface, which renders the PA-decorated SiNPs highly hydrophilic. This leads to their depletion from the fluid interface, forcing its transference to the aqueous phase and, consequently, driving the phase separation of the emulsion. The reduction in palmitic acid concentration in the oil phase drives the system towards a no emulsification region as a result of the PA molecules/SiNPs number ratio.

### 3.3. Stability of Pickering Emulsions in the Water/Decane System

In the previous sections, it has been shown that the interaction of PA with SiNPs enhances the emulsion stability. Thus, it is interesting to study the role of the PA concentration in the Pickering emulsion stabilization. The characterization of the emulsion stability provides information on the quality of the emulsion, for which the relative amount of emulsion in relation to the total volume of the liquid was evaluated. From the above, it is possible to evaluate the stability of emulsions by means of the destabilization index, *DI*, of the emulsion column after aging defined by [31]

$$DI = 1 - \frac{V_e}{V_t},\qquad(3)$$

where $V_e$ is the volume of the emulsion column and $V_t$ correspond to the volume of the entire liquid column. Small values of the destabilization index indicate that the volume of the emulsified column is higher than that corresponding to the destabilized fraction, which in turn indicates a high stability of the emulsion. In the emulsions studied it was found that after the initial destabilization process occurring during the first 24 h, the *DI* reaches a value that can be considered almost stationary. In this work, the degree of destabilization of the emulsions was evaluated one week after their preparation, although a more detailed study showed that the value obtained did not vary significantly during the following 12 weeks. It should be noted that Pickering emulsions are metastable systems, i.e., they do not represent thermodynamic stability. However, the high energy barrier associated with the entrapment of particles at the fluid interface provides kinetic stability to Pickering emulsions which allows one to refer to this type of systems as "ultra-stable emulsions". Figure 5 shows the values of *DI* for emulsions stabilized by palmitic acid decorated SiNPs as a function of the fatty acid concentration obtained in emulsions containing different oil phase volume fractions.

As discussed above, the stability of emulsions with very high and very low values of oil volume fractions have poor stability in agreement with the high values of the destabilization index, which assumes values above 0.8, independently of the palmitic acid concentration. Similar poor stabilities were found for emulsions having low palmitic acid concentrations. Therefore, the values of the destabilization index demonstrated the essential role of the PA molecules/SiNPs number ratio in the control of the stability of Pickering emulsions. In fact, low and high values of such parameters lead to the formation of highly hydrophilic particles, which are depleted from the interface to the aqueous phase, leading to fast destabilization of the emulsions. On the other hand, at intermediate values of the decane volume fraction (in the range 0.2–0.8), the increase in the palmitic acid concentration for fixed values of $\phi_o$ leads to an increase in the emulsion stability, which demonstrates the synergism of particles and palmitic acid on the stabilization of the emulsions, in agreement with the effects observed by Dai et al. [32] for Pickering emulsions stabilized by the interaction of dimethyldodecylamine oxide and silica nanoparticles on the n-heptane/water interface. It should be noted that the destabilization of the emulsions is enhanced within the transitional phase inversion (see the results of $\phi_o = 0.3$), evidencing a higher stability of w/o than o/w emulsions for conditions in which the number of particles is constant. This is explained considering the

effective attachment of highly hydrophilic palmitic acid-decorated nanoparticles to the oil/water interface is hindered, and hence they are easily squeezed out from the interface, leading to a fast destabilization of the emulsions.

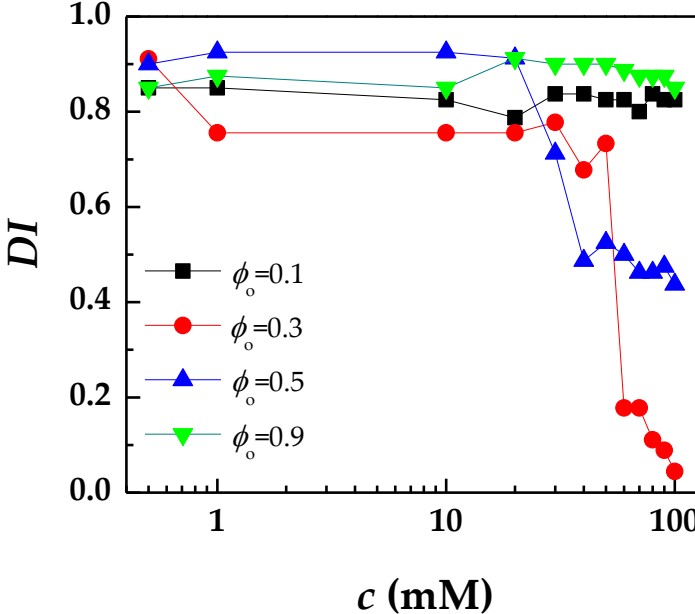

**Figure 5.** Dependence of the destabilization index, *DI*, on the palmitic acid concentration, *c*, for Pickering emulsions having different volume fractions of liquid phases and a fixed particle concentration in the aqueous phase of 1% w/w. The destabilization indices of all the emulsions were evaluated after one week of their preparation.

On the basis of the results, it is clear that the interaction of silicon dioxide particles and palmitic acid is essential for preparing emulsions with enhanced stability. In fact, the creaming of emulsions stabilized only by silica particles occurs rapidly within the first hour after preparation. This can be understood by considering that the raw particles are highly hydrophilic and negatively charged, and hence their adsorption to the oil/water interface is very unstable. On the other hand, palmitic acid allows stabilizing emulsions, especially at relatively high concentrations. Nevertheless, the values of the destabilization indices of fatty acid-stabilized emulsions are always higher than for the Pickering emulsions stabilized by fatty-acid decorated particles with similar palmitic acid concentrations. Furthermore, palmitic acid stabilized emulsions undergo a complete phase separation in less than a week independent of the concentration, whereas when PA interacts with particles, the emulsion column remains unchanged after the initial destabilization occurring with the first 24 h after its preparation. The results in Figure 5 suggest that the catastrophic phase inversion also affects the stability of the Pickering emulsions. Figure 6 shows the evolution of the destabilization index within two compositional lines showing catastrophic phase inversion.

The results show that the oil volume fraction is an essential factor in controlling the stability and type of the emulsions. It can be observed that for o/w and w/o emulsions the value of the destabilization index value decreases as the composition of the emulsions approaches the point of catastrophic inversion, i.e., the stability of the emulsion increases, regardless of the type of emulsion. This behavior is visible for high and low concentrations of palmitic acid in the oil phase, indicating that there is a significant increase in the stabilization of the emulsion by slightly varying the ratio between the volume fractions of the liquid phases [30]. On the basis of the obtained results, it is possible to assume that the catastrophic phase inversion observed appears within a very narrow range of the value of the oil volume fraction (0.2–0.5).

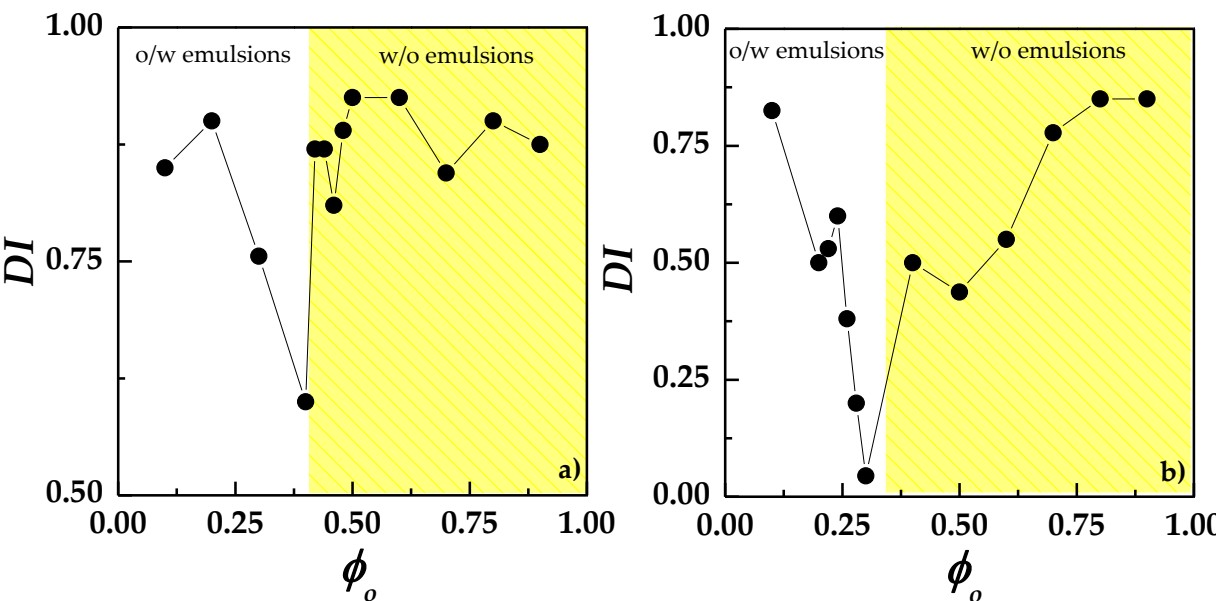

**Figure 6.** Destabilization index dependence on the decane volume fraction ($\phi_o$) for Pickering emulsions with a fixed concentration of silicon dioxide nanoparticles (1% w/w) in the aqueous phase and two different palmitic acid concentrations ($c$): (**a**) 1 mM. (**b**) 100 mM. Notice that the results correspond to two different compositional lines of catastrophic phase inversion.

### 3.4. Droplet Characterization for Pickering Emulsions in the Water/Decane System

Microscopy was used for evaluating the influence of the transitional and catastrophic phase inversions on the size of the droplet. Figure 7 shows a set of images corresponding to Pickering emulsions obtained at a fixed decane volume fraction ($\phi_o = 0.3$) and different alkane volume fractions, as well as the average size (diameter) of the droplet obtained by statistical analysis within the considered compositional line of fixed oil volume fraction. This compositional line corresponds to a transitional inversion.

The results show that independent of the Pickering emulsion composition, the droplets present a high degree of polydispersity. The images obtained for emulsions with palmitic acid concentrations below 60 mM correspond to o/w emulsions, and those obtained for emulsions above 60 mM correspond to w/o. The average size of the droplet remains relatively constant with the palmitic acid concentration within the o/w region, and then the results suggest a slight increase in the average droplet size once the transitional phase inversion is overcome. This may be a result of the interaction between the hydrophobic tails of the palmitic acid molecules decorating the particle surface which can force a partial droplet coalescence or Ostwald ripening phenomena, and an increase in the average size of the droplets [10,33].

The evolution of the size of the emulsion droplets was also evaluated within catastrophic phase inversions. Figure 8 shows a set of images corresponding to Pickering emulsions obtained at a fixed palmitic acid concentration (10 mM) and different decane volume fractions, as well as the average size (diameter) of the droplet obtained by statistical analysis within the considered compositional line of fixed palmitic acid concentration.

The results showed that the average emulsion droplet size undergoes a slight increase within the considered compositional line. This may be understood by considering that the increase in the oil volume fraction results in a decrease in the total number of available particles. Thus, the reduction in the amount of particles results in a reduction in the maximum area that can be covered, and hence the formation of larger Pickering emulsion droplets with a small total interface area is favored to maximize the surface coverage of the droplets in agreement with the finding by Yang et al. [34] for Pickering emulsions stabilized by solid particles of soy protein and chitosan.

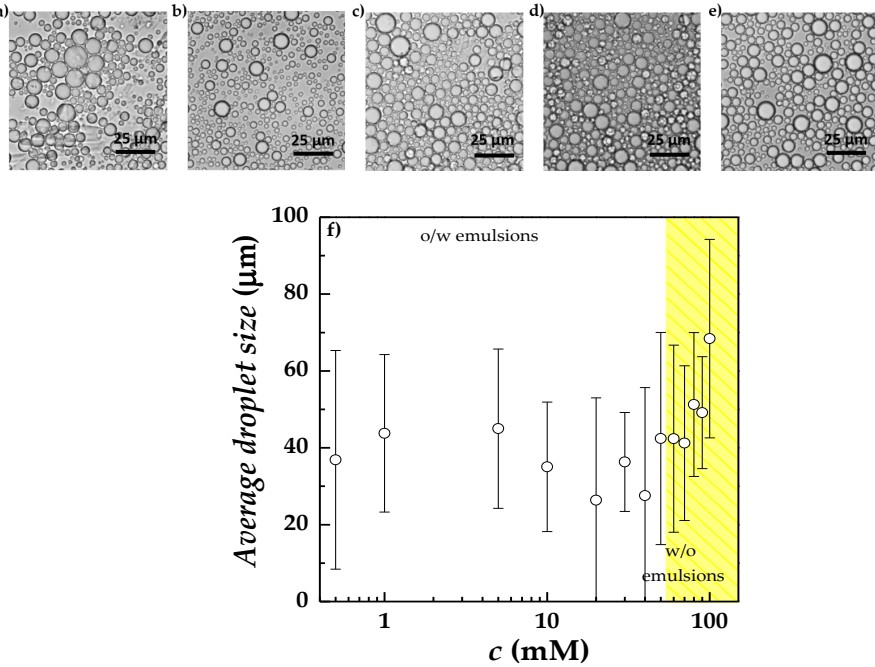

**Figure 7.** Set of images for Pickering emulsions with a fixed concentration of silicon dioxide nanoparticles (1% w/w) in the aqueous phase and different palmitic acid concentrations (*c*) at a fixed $\phi_o = 0.3$: (**a**) 10 mM. (**b**) 30 mM. (**c**) 60 mM. (**d**) 80 mM. (**e**) 100 mM. (**f**) Average size of the droplets of the emulsions of the compositional line corresponding to $\phi_o = 0.3$. The error bars are statistical representations of the polydispersity of the samples.

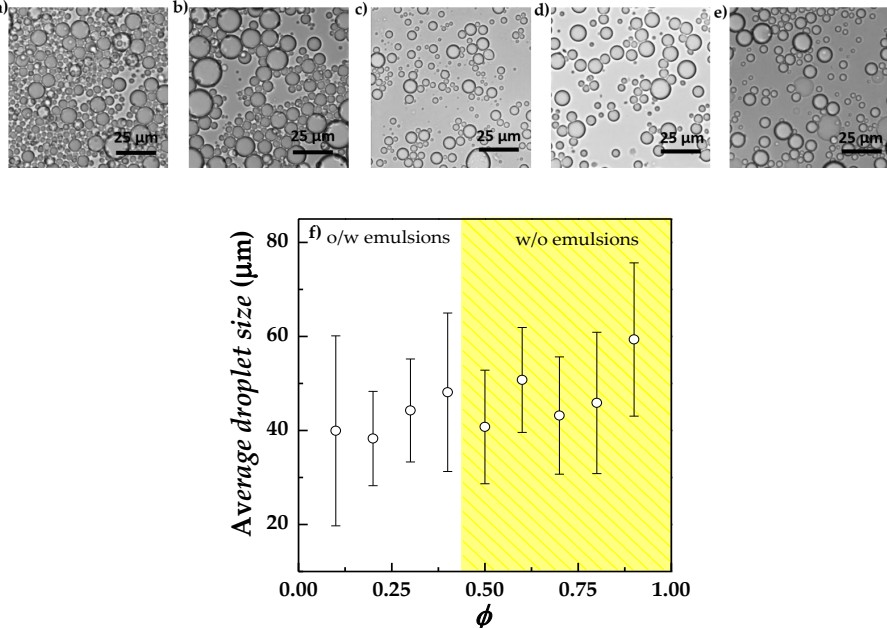

**Figure 8.** Set of images for Pickering emulsions with a fixed concentration of silicon dioxide nanoparticles (1% w/w) in the aqueous phase and different decane volume fraction at a fixed palmitic acid concentration (*c*) of 10 mM: (**a**) $\phi_o = 0.1$. (**b**) $\phi_o = 0.4$. (**c**) $\phi_o = 0.5$. (**d**) $\phi_o = 0.6$. (**e**) $\phi_o = 0.9$. (**f**) Average size of the droplets of the emulsions of the compositional line corresponding to palmitic acid concentration of 10 mM. The error bars are statistical representations of the polydispersity of the samples.

### 3.5. Controlling the Phase Behavior of Pickering Emulsions by the Oil Phase Nature

In the previous sections, the PA molecules/SiNPs number ratio was exploited for controlling the stabilization of the Pickering emulsions and their nature. However, the nature of the Pickering emulsions may also be tuned by modifying other parameters, e.g., the nature of the oil phase [35]. This may provide an alternative strategy for controlling the trapping of the particles at the alkane/water interface by modifying their affinity for the oil phase. Figure 9 shows the simplified two-dimensional phase diagrams obtained for systems having different alkanes (hexane, decane, dodecane and tetradecane) as the oil phase as a function of the oil volume fraction ($\phi_o$) and the initial concentration of palmitic acid ($c$) in the oil phase.

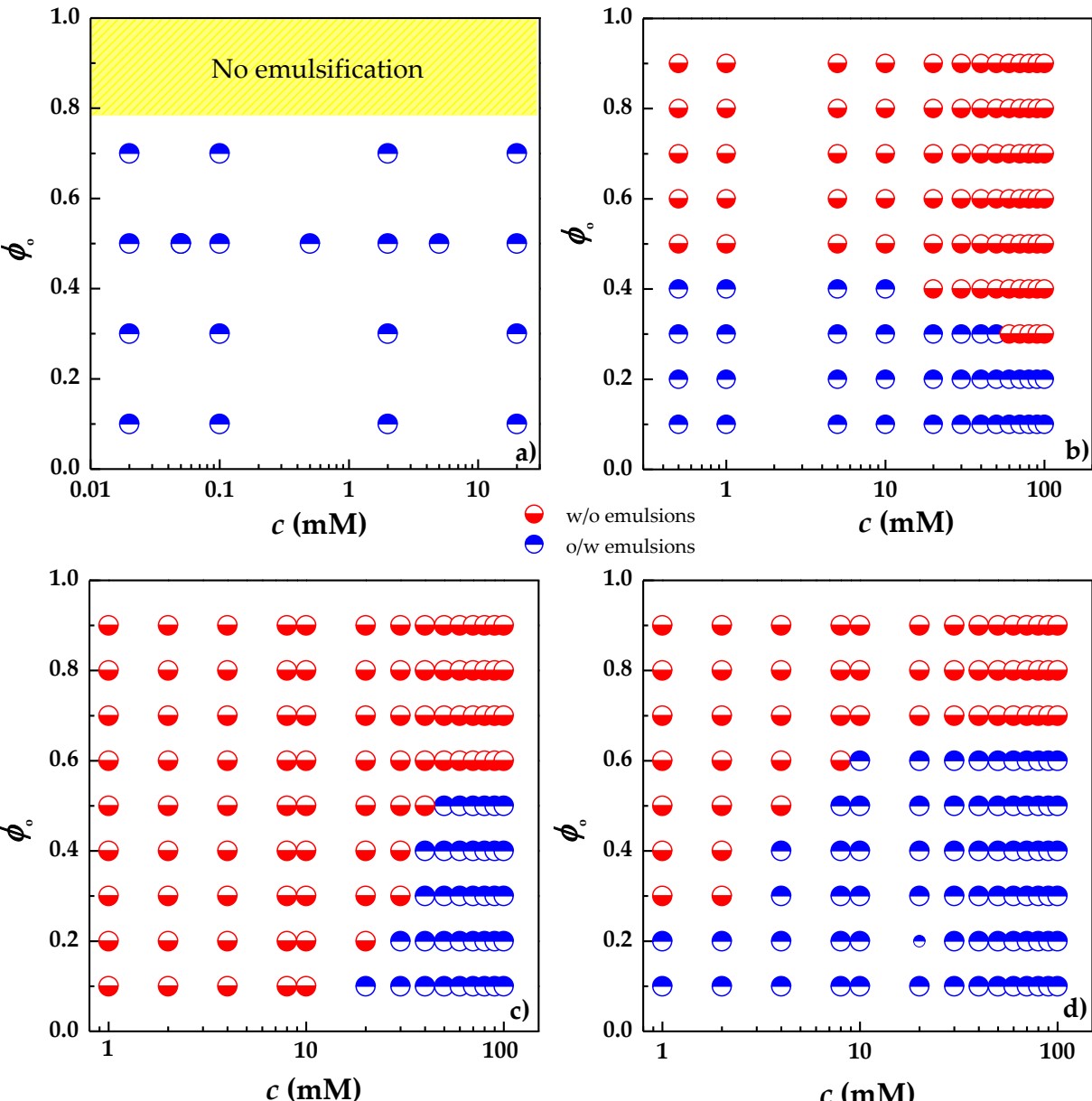

**Figure 9.** Simplified phase diagrams for emulsions stabilized by PA-decorated SiNPs represented in terms of the palmitic acid concentration ($c$) and the oil volume fraction ($\phi_o$). The different regions in which o/w and w/o emulsions appear are shown with different types of symbols. The emulsions were stabilized by a fixed concentration of particles in the aqueous phase of 1% w/w, and different alkanes as oil phase: (**a**) hexane, (**b**) decane. (**c**) dodecane. (**d**) tetradecane.

The simplified phase diagrams provide evidence of strong differences in the emulsion phase behavior depending on the specific alkane used as oil phase. This may be mainly related to the different solubility of palmitic acid molecules in the alkanes, i.e., the hydrophobic mismatch between the length of the alkyl chain of the palmitic acid and that of the alkane. Thus, the solubility of palmitic acid in hexane is expected to be lower than in decane, dodecane and tetradecane. Therefore, for emulsions obtained using alkanes larger than hexane as oil phase, it is possible to enlarge the studied concentration range and, consequently, to enlarge the range of accessible values of the PA molecules/SiNPs number ratios. This allows obtaining decorated particles with different degrees of hydrophobicity, which affects the formation and stabilization of different kinds of emulsions in comparison with the system containing hexane as oil phase [36]. Differences are also evidenced when the phase behavior of emulsions using decane as the oil phase are compared with those containing dodecane and tetradecane.

The results indicate that the increase in the number of carbons in the oil phase shift the phase behavior for a fixed volume fraction of the oil phase to lower values of the PA molecules/SiNPs number ratio, i.e., the inversion between the different emulsion types emerges at lower concentrations of palmitic acid. Thus, for Pickering emulsions having as oil phase hexane, or decane at the lowest values of the PA molecules/SiNPs number ratio, the decorated particles behave as hydrophilic, remaining more wetted for the water, and hence the interface is forced to curve to entrap oil droplets, forming o/w emulsions. This can be understood by considering that despite particles being decorated with the fatty acid, the coverage is not high enough to form a steric barrier allowing water molecules to interact with free silanol groups on the surface of the silicon dioxide particles, rendering them hydrophilic. The situation changes as the PA molecules/SiNPs number ratio increases for emulsions with decane. This is the result of the increase in the coverage of the particle surface by fatty acid molecules, which increases the particle hydrophobicity, pushing the particles towards the oil phase and inducing a curvature of the interface to form water droplets in a decane continuous phase, i.e., w/o emulsions. This situation is not found for emulsions with hexane due to two different factors. On one hand, the solubility of palmitic acid in hexane is lower than in alkanes of longer chains, which in turn reduces the maximum value of the PA molecules/SiNPs number ratio that can be reached, and consequently the maximum degree of hydrophobicity of the particles. This favors the formation of o/w emulsions. On the other hand, the hydrophobic mismatch between the alkyl tail of the fatty acid and the hexane is poor. This reduces the strength of the hydrophobic interactions between the alkane and the fatty acid, favoring the hydrogen bonds of the water with the silanol of the nanoparticles which contributes to the hydrophilic character of the particles [37].

The complex interplay between the surface coverage and the alkane-alkyl tail of palmitic acid hydrophobic mismatch is confirmed for emulsions having dodecane and tetradecane as oil phases. For the case of Pickering emulsions in the system dodecane/water, w/o emulsions emerge for the lowest values of the PA molecules/SiNPs number ratio. This can be understood considering the above-mentioned hydrophobic mismatch. Thus, the enhanced solubility of the palmitic acid with the increase in the length of the alkane chain favors hydrophobic interactions between the hydrophobic tails of the palmitic acid and the alkanes in such a way that the packing density of hydrophobic tails on the particle surface becomes higher than that expected for the single adsorption of the fatty acid. This enhances the hydrophobic character of the decorated particles favoring the formation of w/o emulsions, even though the surface coverage of the particles by fatty acid is relatively low [38]. The transition to o/w emulsions as the PA molecules/SiNPs number ratio increases can be understood by considering the above picture. Thus, the enhanced coverage associated with the incorporation of the alkane between the fatty acid chains can favor the formation of the bilayer, even for relatively low concentrations of fatty acid, leading to a phase inversion.

The phase diagram corresponding to emulsions of the tetradecane/water system confirms the picture proposed for the dodecane/water system. In fact, the stronger hydrophobic mismatch between the alkyl chain of the fatty acid and the dodecane shifts the phase transition to the o/w interface for relatively low values of the palmitic acid concentration. Therefore, on the basis of the above results it is clear that the nature of the oil allows for tuning of the nature of the obtained Pickering emulsions. In fact, under specific conditions in which the palmitic acid concentration and the oil volume fraction are fixed, the oil can drive the system through a double inversion phenomenon. To the best of our knowledge this type of double phase inversion governed by the change of the oil nature does not appear to have been previously reported.

### 3.6. Effect of the Oil Phase on the Stability of the Pickering Emulsions

The previous sections showed that the interaction of palmitic acid with silica nanoparticles enhances the emulsion stability. Thus, it is interesting to study the role of the palmitic acid concentration in the Pickering emulsion stabilization for emulsions with different oils. For this purpose, the evolution of the destabilization index with the palmitic acid concentration has been studied for emulsions with different alkanes as oil phase at $\phi_o = 0.5$ (see Figure 10).

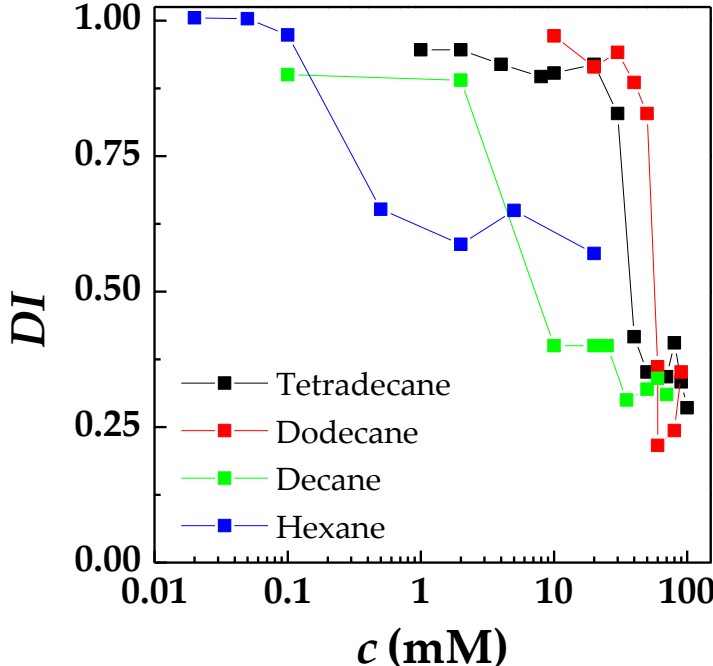

**Figure 10.** Dependence of the destabilization index of the emulsion column on the palmitic acid concentration for Pickering emulsions using different alkanes as oil phases at a fixed $\phi_o = 0.5$. The results correspond to emulsions with an initial concentration of particles in the aqueous phases of 1% w/w.

Again, the stability of the emulsions is strongly affected by the value of $c$ independently of the alkane used. Furthermore, the increase in the length of the alkane shifts the curves of the destabilization index to higher values of the palmitic acid concentration. This may be ascribed to the differences in the nature of the emulsions emerging through changing the alkane used as the oil phase for a fixed value of the palmitic acid. Thus, the modification of the affinity of the particles for the interface as the hydrophobic mismatch between the alkane and the hydrophobic tail of the palmitic acid is changed allows modulation of the expulsion of the particles from the interface, and the strength of the shell surrounding the alkane droplets. On the other hand, the maximum stability of Pickering emulsions

undergoes a two-fold increase by passing from the use of hexane as oil phase to the use of longer chain lengths.

## 4. Conclusions

The present study has been focused on the formation and stability of water/alkane and alkane/water emulsions stabilized by palmitic acid decorated silicon dioxide nanoparticles. The results have shown that silicon dioxide nanoparticles and palmitic acid can interact synergistically during the formation of the drop/continuous phase interface, allowing for emulsion stabilization due to the in situ hydrophobization of the nanoparticles. The Pickering emulsions stabilized by the palmitic acid-decorated silicon dioxide nanoparticles present larger long-term stability than the corresponding conventional emulsions stabilized exclusively by the fatty acid or the nanoparticles. This is due to the formation of a rigid shell around the drops which prevents their coalescence. The type of emulsion (o/w and w/o) can be easily tuned by modifying the PA molecules/ SiNPs number ratio and the nature of the oil phase. The former parameter can be changed either by the modification of the fatty acid concentration at a fixed $\phi_o$ or by the modification of $\phi_o$ at a fixed value of $c$. The variation of the different parameters controlling the emulsion type (volume fraction of fluid phases and palmitic acid concentration) have allowed for the design of a phase diagram of the system. The hydrophobic mismatch between the length of the alkyl chain of the palmitic acid and of the alkane has been revealed as a promising way to modify the nature of the emulsion that can be explained on the bases of the solubility differences of PA in different alkanes.

Future developments of the present work will need deeper investigation of the role of the affinity of the fatty acid for the oil in order to establish better control of the emulsion type. This could represent an important aspect, since in most cases o/w emulsions can be more useful from a technological point of view because the fabrication of emulsions with a hydrophobic core can allow the solubilization and distribution of species of reduced availability.

**Author Contributions:** Conceptualization, R.G.R., E.G. and E.S.; methodology, E.G., A.G.-G., N.S.-A., E.S., J.L.R.-V. and C.C.; software, E.G., A.G.-G., N.S.-A., E.S., J.L.R.-V. and C.C.; validation, E.G., E.S., L.L. and R.G.R.; formal analysis, E.G. and E.S.; investigation, E.G., A.G.-G., N.S.-A., E.S., J.L.R.-V., C.C., F.R., L.L., F.O. and R.G.R.; resources, F.R., L.L., F.O. and R.G.R.; data curation, E.G.; writing—original draft preparation, E.G.; writing—review and editing, E.G., A.G.-G., N.S.-A., E.S., J.L.R.-V., C.C., F.R., L.L., F.O. and R.G.R.; visualization, E.G.; supervision, E.G., L.L. and R.G.R.; project administration, L.L. and R.G.R.; funding acquisition, E.G., F.R., L.L., F.O. and R.G.R. All authors have read and agreed to the published version of the manuscript.

**Funding:** This work was funded by MICINN under grant PID2019-106557GB-C21, and by E.U. on the framework of the European Innovative Training Network-Marie Sklodowska-Curie Action nanoPaInt (grant agreement 955612).

**Institutional Review Board Statement:** Not applicable.

**Informed Consent Statement:** Not applicable.

**Data Availability Statement:** Data available upon request.

**Conflicts of Interest:** The authors declare no conflict of interest. The funders had no role in the design of the study; in the collection, analyses, or interpretation of data; in the writing of the manuscript, or in the decision to publish the results.

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
