# Peer review of "Effects of Oil Phase on the Inversion of Pickering Emulsions Stabilized by Palmitic Acid Decorated Silica Nanoparticles"

_colloids, doi:10.3390/colloids6020027_

Round 1

Reviewer 1 Report

excellent paper which gives an in-depth analysis on the mechanisms involved in the stability of Pickering emulsions. I noted just a few minor points to correct:

1) be carefull with the volumic fraction Φ, it is written sometimes in uppercase, sometimes in lowercase, it would be necessary to standardize

2) regarding the "drop" test described in section 3.2 to identify the nature of emulsions (w/o or o/w), it is noteworthy that it could be difficult to clearly identify the presence of multiple emulsions by this way (w/o/w or o/w/o), especially close to the inversion point. I imagine that you have double checked with optical microscopy, so please precise this point in the text.

3) page 8, last sentence: replace the point by a comma after (in the range 0.2-0.8),

4) page 9, line 302: "It should be noted that the destabilization of drops significantly increases..."

5) page 9, line 315: for me, the DI of fatty acid stabilized emulsions are always higher (not lower) than for the Pickering emulsions... (they are less stable)

6) page 11, line 365: you mention the volumic fraction instead of the concentration (10 mM), please correct.

7) page 11, line 376: The results showed (suppress suggest).

8) page 12, section 3.5: phase instead of pase.

9) page 13, line 437: ...emulsions having dodecane and tetradecane as oil phases.

10) page 14, line 474: suppress it in "...it is strongly affected..."

11) page 15, line 488: The results have shown...

12) Finally, you never mention or discuss the results regarding the IFT of the system, especially regarding the formation of the emulsion at the early stage of the process, because I suppose that it strongly depends on the palmitic acid concentration... Is it possible to include a discussion on this point?

Author Response

Reviewer 1:

excellent paper which gives an in-depth analysis on the mechanisms involved in the stability of Pickering emulsions. I noted just a few minor points to correct:

  • be carefull with the volumic fraction Φ, it is written sometimes in uppercase, sometimes in lowercase, it would be necessary to standardize

Following the suggestion of the reviewer, we have standardized the nomenclature within the whole manuscript.

  • regarding the "drop" test described in section 3.2 to identify the nature of emulsions (w/o or o/w), it is noteworthy that it could be difficult to clearly identify the presence of multiple emulsions by this way (w/o/w or o/w/o), especially close to the inversion point. I imagine that you have double checked with optical microscopy, so please precise this point in the text.

We agree with the comment of the reviewer. Actually, the droplet test provides information related to the type of the external phase existing in the emulsion, and but it does not allow obtaining any information about the nature the internal region, making difficult to extract conclusions about the possible existence of multiple emulsions. However, in our case microscopy imaging have confirmed the absence of multiple emulsions. We have introduced a comment in the test.

  • page 8, last sentence: replace the point by a comma after (in the range 0.2-0.8),

We have corrected in the manuscript.

  • page 9, line 302: "It should be noted that the destabilization of drops significantly increases..."

We have corrected in the text.

  • page 9, line 315: for me, the DI of fatty acid stabilized emulsions are always higher (not lower) than for the Pickering emulsions... (they are less stable)

We have corrected in the text.

  • page 11, line 365: you mention the volumic fraction instead of the concentration (10 mM), please correct.

We have corrected in the text.

  • page 11, line 376: The results showed (suppress suggest).

We have corrected in the text

  • page 12, section 3.5: phase instead of pase.

We have corrected in the text.

  • page 13, line 437: ...emulsions having dodecane and tetradecane as oil phases.

We have corrected the text.

  • page 14, line 474: suppress it in "...it is strongly affected..."

We have corrected the text.

  • page 15, line 488: The results have shown...

We have corrected the text.

  • Finally, you never mention or discuss the results regarding the IFT of the system, especially regarding the formation of the emulsion at the early stage of the process, because I suppose that it strongly depends on the palmitic acid concentration... Is it possible to include a discussion on this point?

Following the reviewer recommendation, we have included a comment about the interfacial tension.

We thank to the reviewer for the comments, they were very helpful for improving our manuscript.

Reviewer 2 Report

This manuscript is well written and hence can be accepted in its present state. 

Author Response

This manuscript is well written and hence can be accepted in its present state. 

We thank to the reviewer for the comment, it stimulates us to continue working in this line.

Reviewer 3 Report

This work was designed to understand the role of the affinity of the hydrophobic chain of a fatty acid (palmitic acid) and the alkyl chain of the oil phase on the stabilization of the Pickering emulsions. The conceptualization of article is very well and article is well written. Also, figures are easily understandable.  

In my opinion, the manuscript could be accepted for publication after making some corrections mentioned below:

  1. Line 85: It could be better to mention product’s supplier in parenthesis such as Zetasizer Nano ZS (Malvern Instruments, UK).
  2. Line 95: Please add supplier’s country.
  3. Line 131: Please use longer instead of larger
  4. Line 256: Please change an with and
  5. Line 362: Authors could add some references about this phenomenon.
  6. Line 384: Please change pase as phase
  7. As a recommendation; It could be better to give some references from other studies in results section such as different ratios, different oils etc. Maybe authors could compare the best results with other studies if possible.

Author Response

This work was designed to understand the role of the affinity of the hydrophobic chain of a fatty acid (palmitic acid) and the alkyl chain of the oil phase on the stabilization of the Pickering emulsions. The conceptualization of article is very well and article is well written. Also, figures are easily understandable.  

In my opinion, the manuscript could be accepted for publication after making some corrections mentioned below:

1. Line 85: It could be better to mention product’s supplier in parenthesis such as Zetasizer Nano ZS (Malvern Instruments, UK).

We have modified the text for a better understanding.

2. Line 95: Please add supplier’s country.

We have added the information.

3. Line 131: Please use longer instead of larger

We have corrected in the text.

4. Line 256: Please change an with and

We have corrected in the text.

5. Line 362: Authors could add some references about this phenomenon.

We have added some references.

6. Line 384: Please change pase as phase

We have corrected in the text.

7. As a recommendation; It could be better to give some references from other studies in results section such as different ratios, different oils etc. Maybe authors could compare the best results with other studies if possible.

We thank to the reviewer for the comments. It is a very interesting point. Unfortunately, this is not possible. All the possible comparisons are included in the text. Up to the date, there are no systematic studies comparing different variables. Therefore, we cannot add more comparion.

We thank to the reviewer for the comments, they were very useful for improving the quality of our manuscript.